# It Is Leadership, but (Maybe) Not as You Know It: Advocating for a Diversity Paradigm in Sports Leadership and Beyond

**DOI:** 10.3390/bs14100860

**Published:** 2024-09-24

**Authors:** Tania Cassidy, Gary Byrne

**Affiliations:** 1School of Physical Education, Sport and Exercise Sciences, University of Otago, Dunedin 9016, New Zealand; 2Independent Researcher, Cork, Ireland; garyleobyrne@gmail.com

**Keywords:** cultural competence, diversity, sports coaching, social identity, organisational behaviour

## Abstract

The need to ‘rethink leadership’ is on the radar of many, from global finance and auditing organisations (e.g., Deloitte) and global sports organisations (e.g., the International Olympic Committee) to national and local sports organisations concerned about the decreasing numbers of participants or the lack of women coaches. Yet, is the dominant Western leadership orthodoxy fit for purpose in the 21st century? The purpose of this article is two-fold. First, to advocate for ways of ‘rethinking leadership’ that challenge the current dominant ethnocentric, gender-biased, leader-centric orthodoxy. Second, to introduce an expanded global and diverse leadership paradigm that is underpinned by clearly delineated dimensions of diversity and cultural competence, which recognises the importance of the organisational and cultural contexts. The literature discussed in this article draws from leadership studies generally and sports leadership and sports coaching more specifically. Key to this article is the discussion of the implications of adopting a diverse leadership paradigm for policy, practice, development, and research of leadership. This advocacy article does not end with a definitive conclusion but rather with an invitation to participate in a journey to realise the potential of diverse leadership.

## 1. Introduction

The need to ‘rethink leadership’ is on the radar of many, from global finance and auditing organisations (e.g., Deloitte) and global sports organisations (e.g., the International Olympic Committee) to national and local sports organisations concerned about the decreasing numbers of participants or the lack of women coaches. The authors of the 2019 Deloitte’s Global Human Capital Trends Report—*Leading the social enterprise: Reinvent with a human focus* [1]—drew on the findings of nearly 10,000 global survey respondents to make a bold ‘call to action’. Namely, it is not the time to tinker at the edges of your organisation; it is time to reinvent it. Interestingly, 80% of the survey respondents held the view that leadership was an important or very important issue, and ‘80% of respondents said that “21st-century leaders” face unique and new requirements’ (p. 6). The authors of the report went on to state that the process of reinvention requires organisations to move ‘beyond mission statements’ and instead ‘*learn to lead* the social enterprise—and reinvent themselves around a human focus’ (p. 2). Moreover, the authors contend that the reinvention process should assist organisations to develop skills and measure ‘leadership in ways that help leaders effectively navigate greater ambiguity, take charge of rapid change, and engage with external and internal stakeholders’ (p. 37). Despite making this ‘call to action’ (p. 80), they acknowledge that reinventing organisations is not easy because ‘while organizations **expect** new leadership capabilities, they are still largely promoting traditional models and mindsets’ [1] (p. 37, **emphasis in the original**).

A year prior to the publication of the Deloitte Report [1], Chin and colleagues [2] made similar observations. They note that much of the dominant leadership practices, as discussed in the English literature at least, tend to reflect ‘an Anglo-Western-male capitalistic perspective’ (p. 323). This led them to contend that if leadership practices are to be relevant and beneficial for our increasingly diverse communities, then orthodox views of leadership need to be challenged. One possible way to challenge ‘idealised’ Western leader-centric models and potentially operationalise Deloitte’s ‘call to action’ [1] is seen in the earlier work of Chin and Trimble [3], where they specifically focus on diversity and leadership. They suggest, amongst other things, there is value in the following:Providing conceptual clarity as to what constitutes ‘diversity’ (i.e., inclusion, cultural competence, equity) and ‘cultural competence’ (i.e., climate, core/curriculum, composition) within organisational behaviour;Situating discussion of diverse leaders and diverse members within organisational and cultural contexts, allowing for a multilevel analysis of leadership, which incorporates leaders, members, units, groups, organisations, and systems;Utilising an alternative sociocultural paradigm of leadership that encompasses social identity, decolonial, and critical leadership approaches.

This article builds on our previous work [4,5,6,7] by being more explicit about diversity, cultural competence, and the centrality of the organisation in diverse leadership. We openly acknowledge leadership is, amongst other things, a ‘complex, social, cultural, relational, situated, and pedagogical practice’ [6] (p. 127), but we also accept that it is not possible to ignore the behaviours of individuals. As such, we contend there is value in being explicitly aware of your behaviour or the behaviours of others because this knowledge can then be a precursor for asking questions about ‘where, how and why leadership work is organised and accomplished’ [8] (p. 196).

It is the above work and associated theories and research that we draw on to contribute to this Special Issue, *Rethinking Leadership Development*. To begin the ‘rethinking’ process, there may be some value in reflecting on the title of the Special Issue. Does its title conjure up images of a bold ‘call to action’ [1] or more of a tweak of the status quo? Spiller and colleagues [9] have been bold when discussing leadership and supporting practitioners on their ‘*developmental leadership journey*’ (p. 3, *emphasis added*). The analogy of being on a journey suggests an ongoing process rather than a definitive outcome, and, for some, at least, a journey conjures up feelings of excitement and exploration. Viewing leadership as an ongoing process may be advantageous in the modern world because leadership in contemporary times often 

“requires stepping into the unknown, developing sharper powers of observation, being more comfortable with uncertainty, and finding new and better ways to tackle situations, relying not only on rational thinking but also on the much broader sets of intelligence with which each of us is endowed [9] (p. 3).”

The context of our discussion of leadership in this article is sports leadership and leadership in sports coaching (see [6]). Focusing on leadership in sports is a topical issue. For example, at the beginning of the Paris 2024 Olympic Games, the International Olympic Committee posted the following:

“Paris 2024 is the first Olympic Games in history with full gender parity on the field of play, thanks to the distribution of an equal number of quota places to female and male athletes by the International Olympic Committee (IOC) [10].”

At first glance, these figures appear to be a win for gender equity in the context of athletes attending the Olympic Games. However, if we examine the context of this parity in terms of the gender diversity of sports leaders, which includes sports coaches who develop and support the athletes, it is not so positive. Burton and Leberman illustrated this by highlighting the number of women in leadership positions in Olympic-related sports organisations. In 2017, 

“[m]en ran 33 of the 35 international federations affiliated with the Olympics… Only 5.7% of International Federation presidents were women, 12.2% were vice presidents, 13.1% were executive committee members, and only 24.4% of the IOC members were women. More concerning, perhaps, is the fact that a number of international federations have no women on their executive committees despite having high levels of participation by women… At a national level, the figures are not much better. Only 9% (389/4303) of national presidents across the world were women (p. 2) [6] (p. 134).”

The figures are no better for women coaching at the Olympic Games. The average percentage of accredited female coaches participating in the Olympics over the past 14 years has hovered around the low double figures. At the Winter Olympics in Vancouver in 2010, 10% of the accredited coaches were women [11], while at the Summer Olympics in Tokyo 2020, 13% of the accredited coaches were women [12]. While those figures make for sorry reading, figures for some countries are even worse. For example, at the Summer Olympics in London 2012, only 4% of the accredited New Zealand coaches were women (compared to the international average of 11%) [12], while in Tokyo 2020, 6% of the accredited New Zealand coaches were women (compared to the international average of 13%) [13].

Women are not the only under-represented group in leadership roles in sports; non-European men are also under-represented. For example, in the elite rugby union, men who identify as Pasifika make up 38% of players who are contracted to the All Blacks or All Blacks Sevens between 2021 and 2023 [14]. Despite the large number of Pasifika men who play rugby union at the elite level, there is a paucity of Pasifika men coaching or holding other leadership roles at the top level of rugby union. Similarly, the English Premier League [EPL] football showcases a diverse playing population, yet there have only been 11 non-European EPL managers in over 30 years [15].

The purpose of this article is to advocate for ways of ‘rethinking leadership’ that challenge the current dominant ethnocentric, gender-biased, leader-centric orthodoxy. We draw on the theoretical discussions of Chin and Trimble [3] to advocate for an expanded global and diverse leadership paradigm that is underpinned by clearly delineated dimensions of diversity and cultural competence and which recognise the importance of the organisational and cultural contexts. Specifically, we advocate for recognising the value of understanding why and how diversity and cultural competence can be integrated into sports leadership, as well as recognising that the leaders and members of the organisation have multi-layered and overlapping social identities. This advocacy work occurs by exploring a deep-level ‘global and culturally diverse leadership’ paradigm [3].

This article is significant for four reasons, with the first and second reasons being interlinked. This article is topical and contributes to what researchers in the sports leadership and sports coaching fields have called a key challenge, one of cultural competence [16,17,18]. ‘Cultural competence’, however, remains largely ill-defined and conceptually ‘cloudy’ in these fields [19,20], a critique that can equally be levelled at ‘diversity’ [21]. Thus, this article is significant because it advocates for the integration of diversity into all discussions of leadership. The third reason this article is significant is that asking ‘why’ questions of the leadership orthodoxy ‘raises the possibility that scholarship and leadership practices become more relevant for progressively diverse populations’ [6] (p. 128) and supports those urging leaders to act ethically. The power of asking such questions and expecting leaders to act ethically was highlighted when questions were asked why the now ex-Spanish Football Association chief, Rubiales, thought it was appropriate to give player Hermoso an unsolicited kiss on the lips after Spain’s Women’s World Cup victory in 2023. Fourth, this article contributes to the discussion on leadership by shifting the discussion from primarily focusing on the individual leader–member exchange (e.g., athlete–coach dyad) towards a broader discussion that incorporates sociocultural, pedagogical, and indigenous theories, which focus more on collectivist meaning-making rather than competitive unidirectional leader-centrism.

To achieve the purpose of this article, as described above, it is structured thus. Following this introduction is a discussion of a body of literature that views leadership from a sociocultural perspective and challenges the orthodoxy associated with a Western leader-centric model of leadership. The literature is drawn from leadership studies generally and sports leadership and sports coaching more specifically. Succeeding this discussion is an introduction of a diverse leadership paradigm that challenges the Western leadership orthodoxy by asking new questions and advocates for expanding our thinking about ‘how leadership is perceived, enacted and appraised’ [3] (p. 17). The penultimate section of this article is a discussion of the implications of adopting a diverse leadership paradigm for policy, practice, development, and research of leadership. This advocacy article does not end with a definitive conclusion; rather, it ends with an invitation to participate in a journey to realise the potential of diverse leadership.

## 2. Towards a Sociocultural Paradigm Shift: What Does the Literature Say?

The following is not an exhaustive overview of the leadership literature, which is not possible given the expanse of academic and professional literature published on leadership. For example, when Cassidy and colleagues [6] typed ‘leadership’ into Google Scholar, the outcome was ‘About 4,520,000 results’, with ‘About 779,000 results’ occurring from 2017. In the latter category, 29% of the results discussed leadership from a sociocultural perspective, while the majority were informed by psychology and business/management/organisational studies, with the focus of the latter being on individual human behaviour. In line with Deloitte’s ‘call to action’ [1] and Chin and Trimble’s [3] appeal for leadership to acknowledge diversity, cultural competence as well as organisational and cultural contexts, the following discussion focuses on sociocultural perspectives of leadership which challenge the Western leader-centric models of leadership. Additionally, it draws on recently published discussions of leadership generally and, more specifically, leadership in sports and sports coaching [6].

Cassidy and colleagues [6] describe leadership in sports, when viewed through a sociocultural lens, to be, amongst other things, ‘a complex, social, cultural, relational, situated, and pedagogical practice’ (p. 127). Similarly, Leberman and Burton [22] argue for leadership to be viewed as ‘a “moment” of *social* relation’ (p. 8, *emphasis added*), while Schull [23] advocates for recognition to be given to the way in which perceptions of leadership are ‘influenced by an individual’s experiences—both past and present—and are shaped by collective beliefs within a specific *socio-historic context*’ (p. 103, *emphasis added*). However, calls to view leadership from a sociocultural perspective are not new. Over 20 years ago, Osborn and colleagues [24] made a case for a *contextual* theory of leadership.

In the 21st century, leaders are expected to deal with rapid change and uncertainty whilst developing practices in order to recognise diversity and make their organisations more sustainable [6]. But to achieve the latter, ‘they need to do more than reduce their organisation’s carbon footprint’ [6] (p. 129). Adopting sustainable practices also requires ‘leaders to consider the human capital involved in their organisations and the way in which their organisation is culturally sustaining’ (p. 129). Given the above, it is not surprising that there is a growing demand for knowing *how* to become a leader. While the research and resources to support leaders in understanding leadership from sociocultural perspectives are limited, researchers have been working in this area for some time. For example, nearly 30 years ago, Blunt and Jones [25] explored the limits of Western leadership theory in the context of East Asia and Africa and outlined ‘some preliminary alternatives to Western notions of leadership, in an attempt to show ‘*how*’ culture might be taken into account’ (p. 7, *emphasis added*).

While there may be a desire to focus on how to become a leader who can cope with the challenges of the 21st century, it is also important to recognise that there is ‘no single best way to lead’ and ‘effective leadership is contingent on particular environmental and elemental factors that leaders must recognize to optimize their own leadership attributes’ [9] (p. 13). With this in mind, Spiller and colleagues [9] describe and discuss ‘wayfinding leadership’, which is ‘guided by a philosophy of recognition’ (p. 17). They contend that such a philosophy

“requires leaders to recognise the importance of cultivating the “potential” of people, the existence of “many sources of knowledge”, the need to “continuously refresh” their thinking, the value in being “emotionally self-aware”, the benefits of seeing “not only the ‘thing’ but also the between things… [and] discern relationships and patterns”, and the importance of knowing what is happening in the “*now*” whilst being clear about the destination they wish to go (p. 17, *emphasis in original*).”

Informed by the above philosophy, Spiller and colleagues [9] identify five ‘waypoints’ that they use to frame their wayfinding perspective of leadership. One of the waypoints focuses specifically on an ‘[o]rientation on *how* to lead’ (p. 29, *emphasis added*). The principles of this waypoint encourage leaders to ‘remain open to new possibilities’ (p. 29), understand their world is about ‘becoming’, hold ‘a deep belief in the power of co-constructing one’s destiny’ (p. 36), and place attention ‘on the process’ (p. 38) and ‘the growth in people on the journey’ (p. 41).

One of the reasons why researchers are paying more attention to sociocultural and pedagogical approaches to leadership is that they are recognising a desire or expectation that leaders acknowledge the complexities of leadership. The value of adopting these approaches to leadership is highlighted by Paris [26], who drew on his experience of working in an education context to suggest that there is value in using the term ‘*culturally sustaining pedagogy’.* This term ‘supports the value of our multiethnic and multilingual present and future’ and requires the adoption of pedagogical practices that are ‘responsive of or relevant to the cultural experiences and practices of young people’ [26] (p. 95).

A challenge facing leaders in this increasingly diverse world is that it is unlikely that they will only interact with people who look, think, and behave like them. One strategy that has been suggested to enhance interactions between members of diverse populations has been for people to become more culturally competent. Yet, there have been debates over what constitutes competence as well as cultural competence. The debates occur because there are numerous interpretations/models of competence, with many of them focusing on defining ‘measurable, specific, and objective milestones describing what people have to accomplish to consistently achieve or exceed the goals for their role, team, division, and the whole organisation’ [27] (p. 28). Frelin [28] questioned this interpretation and drew upon the work of Biesta to describe competence as the ‘ability to do things’ (p. 6), all the while recognising that having ability alone is not enough. She contends that the practitioner, in this case, the leader, is required to demonstrate competence, as well as ‘educational judgement’, aka ‘professionality’ [28] because, as Biesta [29] points out, ‘a teacher who possesses all the competences teachers need but who is unable to judge which competence needs to be deployed when, is a useless teacher’ (p. 10). 

Other debates have occurred around what constitutes cultural competence. The term cultural competence evolved from terms such as cultural sensitivity and cultural awareness and has been described as the ability to ‘…apply knowledge and skill appropriately in interactions with clients’ [30] (p. 9). Additionally, the ability to apply knowledge and skill requires recognition and understanding of foundational notions such as culture, race, diversity and marginalisation [30]. Yet, there have been debates around the merits of developing approaches such as cultural competence and associated concepts, e.g., culturally relevant pedagogy [31], culturally responsive teaching [32], culturally relevant education [33], and culturally relevant cycle [34]. One of the challenges levelled at the above approaches is that they can ‘reflect an essentialist view of culture, that is, that there is an “essence” that marks all members of one cultural group as being distinctively different to another’ [6] (p. 40). Suggestions have been made as to how this essentialism can be overcome. For example, Curtis and colleagues [35] advocate for practitioners, in this case, leaders, to ‘critique the “taken for granted” power structures and be prepared to challenge their own culture and cultural systems rather than prioritise becoming “competent” in the cultures of others’ (p. 16). Hammersley [36] challenges essentialist views of culture by engaging with the concept of intersectionality, which recognises the ways in which our identities ‘are multiply defined in terms of “race”/ethnicity, sexual orientation, social class, ability/disability, and other dimensions as well’ (p. 56). Organisations also have a role to play in overcoming such essentialism. McNamee [37] suggests that the pursuit of *eudaimonia* (human well-being or flourishing) is worthwhile but recognises the difficulty with such a position, especially when living in this diverse world, because there is no universal understanding of what constitutes ‘excellence’ or a ‘good life’. Yet, McNamee [37] contends that focusing on developing excellence could be a way to engage with the ideas of competence, specifically by acknowledging that judging competence and excellence is influenced by the context in which it occurs, and therefore, any judgement requires flexibility and ‘professionality’.

### Sports Leadership and Leadership in Sports Coaching

Leadership practices and the theorising of leadership in the sporting context reflect the trends in the broader leadership context, with the majority of the sports leadership research being informed by the management/organisational literature, e.g., [38,39,40], and by psychology, e.g., [41]. Yet, there are hints, even within the aforementioned texts and certainly elsewhere, that sports leadership is on the brink of a paradigm shift. For example, the *Journal of Sport Management* published a Special Issue—*Sport Leadership: A New Generation of Thinking* (2018). The rationale the editors gave for the Special Issue [42] was that there was 

“… a foundational bias in the literature towards researching traits and characteristics of individual leaders (often white, male) (Burton, 2015) where leader-centred perspectives and theories, such as transformational, transactional, and charismatic, have taken prominence (Welty Peachey et al., 2015). A response to concerns about the leader-centric focus (sometimes referred to as the hero leader—often propagated by the sports media) has been the emergence of follower-centred perspectives on leadership (Uhl-Bien, Riggio, Lowe, & Carsten, 2014) (p. 77).”

When calling for contributions to the above Special Issue, the editors requested that contributors emphasise what they called a ‘social construction’ view of leadership, which ‘views leadership as a social, collaborative, relational experience focusing on the idea that leadership emerges from the interactions and constructions of people in a particular context’ (p. 77). It was acknowledged that

“leadership models developed in Western settings are largely based on assumptions that the basis of human motivation is individualism and competition (which is less relevant for many indigenous cultures). It is potentially these assumptions that have fuelled the preoccupation with the individual leader as the focus of leadership scholarship (Grint, 2005) (p. 80).”

The editors also recognise that the diverse contexts and social structures within which leaders operate also impact who gets the opportunities to lead and how they lead. So, the stated mission for the above Special Issue was to ‘understand leadership from the point of view of those less visible within sports management/journals (i.e., women, non-Western societies)’ (p. 80). But the hint of a paradigm shift in sports leadership is not confined to sports management. Work occurring in sports psychology also hints at a conceptual shift that recognises the ‘cultural composition of both the client and the practitioner’ [43] (p. 283; also see [19]). However, Schinke and Moore note that despite sports psychologists often working with athletes from diverse cultural backgrounds, the field of sports psychology has ‘been slow to join the dialogue or to learn from’ professional psychology colleagues who have ‘long embraced the integration of cultural practices’ [43] (p. 283). This lack of engagement was one rationale given for the *Journal of Clinical Sport Psychology* publishing a Special Issue in 2011 entitled *Culturally Informed Sport Psychology.*

In the sports coaching literature, there is a growing body of scholarship that focuses on sociocultural competence, cultural context, cultural diversity, cultural awareness, and diversity and inclusion, e.g., [20,21,44,45,46,47,48]. Additionally, there is a hint that there is a greater willingness by some to entertain a paradigm shift in how leadership is understood and practised. For example, in a study that explored how Olympic athletes viewed the characteristics of their leaders and managers, the athletes identified their least favoured characteristics as being self-focused, haughty self-belief, inauthentic, manipulative, and success-obsessed [49]. These least favoured characteristics are consistent with the ‘idealised’ command-and-control ‘Alpha’ leadership style, eschewed by Trimble (see [50]), which is prevalent in sports leadership and is particularly revered in the West, where sports leaders and managers are reluctant to appear collaborative, altruistic, fluid, or remotely vulnerable in the face of complexity, plurality, and uncertainty (see [51,52]). These pathologies are captured by Gearity, who decries ‘[t]he (sexist, racist, heteronormative) Great Man myth of leadership with its ties to business and military rationalities’ [53] (p. 375) that are so apparent in sports coaching. Gearity explains that these characteristics are supported in neoliberal times because they reflect the view of ‘CEO-coach-as-sovereign’, privileged, self-orientated, and all-powerful (p. 377).

Recently, other sports coaching scholars have argued that there remain deeply ingrained foundational social identity biases such as racio-ethnic (folkloric) prescriptions about who makes an effective leader [44,53]. These biases are foundational to the Western leader-centric paradigm that fetishises the podium culture of winning at all costs as the ultimate (and only) predefined and reductive measure of leadership effectiveness. This ‘idealised’ leadership approach is reproduced by homogenous and dominant in-groups. Recognising this, Culver and colleagues [54] call for improved diversity in coach development at the levels of recruitment, language, learning, and development. Similarly, Rynne and colleagues [55] advocate for coach developer training to enhance understandings of indigenous participants’ local knowledge systems matched by appropriate instruction. This aligns with what Gearity and colleagues [18] regard as coach developers’ racial privilege.

The above-referenced work in sports leadership and sports coaching suggests that researchers in these fields are beginning to acknowledge the need for an ethical, nuanced, multilevel analysis of leadership that incorporates not just individuals, dyads, and teams but larger groups, organisations, and systems within cultural contexts, which views leadership as ‘socially constructed’, i.e., leadership as a social, relational, collaborative, contextual, complex, and strategic dynamic in ever diverse organisational and cultural contexts, e.g., [42,56,57,58,59]. The body of literature supporting this paradigm shift (re)conceptualises sports (coaching) leadership as sociocultural, pedagogical, and indigenous [6]. Additionally, those researchers whose work reflects the ‘social identity’ paradigm in sports (i.e., race, ethnicity, gender, sexuality, (dis)ability, class, lived experience) not only shed light on social justice biases, barriers and underrepresentation, e.g., [60,61,62], but are also beginning to inform the bourgeoning literature on racio-ethnic issues and the gendered nature of sports leadership, gender equity and institutionalised underrepresentation of women in sports leadership, e.g., [59,63,64,65,66]. These interrelated paradigm shifts represent epistemological and normative turns towards more responsible, principled, ethical, altruistic, virtuous, inclusive, and collectivist positions in sports leadership and constituent fields.

Despite the above shifts, much of the discussion of leadership in sports coaching is confusing, and there appear to be assumptions by many that there is a shared understanding of what leadership means. Arguably, this lack of definitional and conceptual coherence [67] is exacerbated by the value coaches place on biographies and autobiographies of winning individual coaches, which Stoszkowski and Collins [68] identify as coaches’ second most popular method used to inform their practice. However, Cassidy and colleagues [6] contend that ‘the relevancy and variability of the quality of the biographies and autobiographies could contribute to the lack of clarity surrounding the discussion of leadership in informal sports coaching resources’ (p. 136). Alongside the confused state of the discussion around leadership, there is also limited conceptual clarity around what constitutes diversity and cultural competence and how these concepts are delineated and operationalised [19,20,21]. While the sociocultural paradigm shifts highlighted above are positive, there remains work to be carried out to understand how sports organisations can leverage differences to become culturally competent. Also, there remains the challenge and opportunity regarding how to bring leadership and diversity together and how to move beyond a narrowly defined leader–follower exchange paradigm.

## 3. Beyond Leader–Member Exchange (LMX): A ‘Diverse Leader Member Organisational Exchange’ (DLMOX) Paradigm [3]

Expanding the entrenched (dyadic and dominant) leader–member exchange approach to sports leadership is a key focus of this article. Building on the introduction and the discussion of the referenced literature and gaps therein, the attention now turns to introducing key ideas from Chin and Trimble’s ‘diverse leader member organisational exchange’ paradigm (DLMOX) [3]. First, it is worth saying that, in this section, we explicate a leadership *paradigm* and not a linear behaviouristic leader trait model with typically discrete specifications. As such, this section is conceptually oriented in keeping with the advocacy impulse behind this article. Second, the DLMOX paradigm is integrative and interdependent, so discussing one aspect as a discrete unit is impossible and unhelpful in any event. Third, because the paradigm is not a prescriptive leadership ‘treatment’ nor a copper-etched system, there is a certain fluidity to how the concepts might apply to the readers’ context.

To begin, it is instructive to briefly map out Chin and Trimble’s underlying logic [3] using the following propositions:Leadership concepts, practice, and research are largely representative of Western (North American, Eurocentric), ethnocentric, and gender biased (white and male) and are narrowly defined in terms of other dimensions of social identity (middle class, heteronormative, able-bodied). This has led to a Western leader-centric paradigm that has become the ‘idealised’ universal approach, and this hegemony has gone largely unchallenged;This leader-centric and techno-rational hegemony is increasingly inadequate in the face of globalisation, shifting social values (e.g., social justice), and proliferating heterogeneity. Culturally competent leadership is required as the catalyst for more diverse, responsible, and responsive leadership;The orthodox ‘leader member exchange’ (LMX) orientation is dyadic and lacks an appreciation of the diversity of social identity and lived experience, as well as lacking a multilevel analysis of diverse organisational and intercultural contexts;Recognising and including diversity (D) and organisational contexts (O) is critical in order to transcend the prevailing leader-centric, dyadic, and decontextualised LMX paradigm;Cultural competence, as a catalyst for diversity, occurs across three organisational dimensions: *climate* (values, norms, policy), *core/curriculum* (strategy, programmes, operations, content, resources), and *composition* (leaders, members, stakeholders);A ‘diverse leader member organisational exchange’ paradigm (DLMOX) is proposed to expand previous conceptualisations of leadership.

### 3.1. Challenging the Western Canon of Idealised Leadership

Chin and Trimble’s point of departure is that ‘a paradigm for diversity leadership is important… [because] we need to ask new questions, create new paradigms, and identify new dimensions to expand our thinking about how leadership is perceived, enacted, and appraised’ [3] (p. 17).

The rationale Trimble and Chin [69] provide for challenging the Western canon of ‘idealised’ leadership is that

“[w]ith few exceptions the current theories and research on leadership neglect the value of cultural diversity when considering different leadership styles. [They] omit dimensions of diversity in researching how leadership is exercised, and the values effective leaders promote. Yet, there’s little question the growing population diversity in our complex and uncertain world requires a deep understanding of how effective leadership is exercised and the role leaders will play in managing the changes ahead… [the] entrenched models are overwhelmingly ethnocentric, and gender biased. They draw on narrow, cultural-specific knowledge and practices that simply are not relevant for a diverse and global population, nor applicable in varying contexts and changing social environments’ (p. 1).”

This rationale is further evident in Chin’s keynote presentation [70], in which she asks questions about the consequences the ‘entrenched models’ have on our understanding of leadership and what they might mean for people and their communities. She asks questions that focus on who and what ‘is congruent with leadership and being a leader’ before reflecting on the ways in which orthodox models of leadership and their subsequent rules are ‘often culturally specific and narrowly based on those who are already in privileged positions’. She continues by pointing out that so often 

“[w]e base our principles, policies, and practices on a culture that reinforces values that are held by a dominant group, which mitigates against change, diversity, inclusion, and collaboration. And diversity is not just about counting the numbers and saying we have representation.”

Chin [70] goes on to say that if we are to challenge the orthodoxy of leadership, then ‘you have to pay attention to those around you; who are the unfavoured category? Effective leaders develop shared outcomes that reflect the diversity and goals of all their people and members’. Having mapped out the general logic and provided the rationale for the DLMOX paradigm [3], the following discussion focuses on the characteristics and the dimensions of diverse leadership, as well as how cultural competence can be viewed as a catalyst for diversity that can be leveraged throughout the organisation.

### 3.2. Characteristics, Dimensions, and Prototypes of Diverse Leadership

Diverse leaders and members in diverse organisations are at the heart of Chin and Trimble’s multilevel DLMOX paradigm [3] as they seek to ‘incorporate the ways diversity shapes our understanding of leadership and its effects’ (p. 6). Instrumental to the development of the DLMOX paradigm were responses to surveys and interviews conducted with 190 culturally diverse leaders from across the globe, who were asked to describe the characteristics that were most and least important to them in their leadership also [69,70,71]. In doing so, the participant leaders identified the following characteristics:*Most important*: adaptability, integrity, authenticity, honesty, and communication;*Least important*: aggressive, conflict-inducer, dominant, self-centred, and status-conscious.

Subsequently, Chin and Trimble [3,70,71] identified five ‘dimensions’ of diverse leadership (i.e., constructive, imaginative, principled, synergistic, virtuous) that underpin their non-Western diverse leadership ‘prototypes’ (i.e., Daoist, Silent/Invisible/Reluctant, Benevolent Maternalism/Paternalism, Feminist, Servant, Theory Z) (also see [2]). Fundamental to all these leadership ‘prototypes’ are mutuality, collectivism, benevolence, familial affiliations, and the self as *inter*dependent. Importantly, diverse leaders operate from an ‘affirmative paradigm’, i.e., have courage (to stand up, risk and disrupt), be resilient, maintain integrity, accept the title of leader, be confident (not apologetic for your difference), and translate negatives into positives. While we have merely touched on the above characteristics, dimensions, and prototypes, we can nonetheless see that diverse leadership is a collective and interdependent endeavour rather than the assertive efforts of the individual leader. However, for Chin and Trimble [3], diversity is more than this; diversity is indeed about inclusion and equity, but at the heart of diverse leadership is cultural competence, which merits some attention.

### 3.3. Cultural Competence as a Catalyst for Diversity: Climate, Core/Curriculum, and Composition

Chin and Trimble [3] suggest that cultural competence is evident when the organisation pays attention to issues of diversity as well as being intentional and systematic about diversity in its *climate* (values, norms, policy), *core/curriculum* (strategy, programmes, operations, content, resources), and *composition* (leaders and members). Cultural competence, therefore, is much more than counting the numbers for representation by obtaining a seat at the table, claiming affirmative action, or ad hoc cultural sensitivity training [70,71]. Cultural competence is what the organisation does. Cultural competence, as a catalyst for diverse leadership, is an important driver behind the DLMOX paradigm because it brings leaders, members, and organisations together as diversity is embedded purposefully in the organisation’s climate, core/curriculum, and composition. These points are developed further in the next section as we consider some implications of shifting towards a diverse leadership paradigm.

## 4. Implications and Discussion

We now consider the implications, i.e., the significance, tensions, and ramifications of a global and diverse leadership paradigm for the policy, practice, development, and research of leadership, and, indeed, how leadership might be (re)conceptualised. The reconceptualisation initially moves from higher levels of abstraction—rethinking leadership, out of necessity, starts at a conceptual level—to more specific implications. These implications are informed, or otherwise inspired, by the work of Chin and Trimble [3] and colleagues [2,72,73], whose work is implicit in the primer questions chosen to frame the discussion in this section. The primer questions stem from an overarching question that can broadly be summarised thus: What might enable diverse leadership, and what might constrain diverse leadership? The following (sub)questions have been identified because of their significance for rethinking leadership development. In answering each (sub)question, we discuss corresponding tension and ramifications because these implications are as pressing for leadership across multiple domains as they are for leadership in sports organisations and sports coaching.

### 4.1. Am I Constrained as a Leader if My Social Identity, at Least as It Appears on the Surface, Is Not so Obviously Diverse?

To answer this question, it is important to recognise two key points. First, ‘lived experience’, as an oft-overlooked dimension of social identity, matters. It matters because diverse lived experiences may contribute to divergent thinking, which is perhaps not as readily identifiable as other dimensions of social identity. Second, culturally competent leadership requires a level of ‘cultural humility’, which Chin [70] describes as other-oriented, curious, and humble. Chin [70] pointedly identifies in her keynote address that ‘if you think you’re culturally competent then you’re not’ because there is an even deeper level of understanding, which is cultural humility, and cultural humility is about ‘adopting an interpersonal stance that is other-oriented’ (also see [19]). Without such cultural humility, tensions may arise if it is implied that unless leaders have easily identifiable diverse social identities, then they cannot expect to be diverse leaders. Also, there is the potential for tensions to arise if certain social identities and certain lived experiences are seen as favoured while others are unfavoured. The ramification of creating favoured and unfavoured categories of social identity is that bias and exclusion continue. To this end, acknowledging that life experience is also part of one’s social identity, alongside other dimensions of social identity and their intersectionality, enables diverse leadership and mitigates exclusion and inequity. The next (sub)question of significance is a related one.

### 4.2. With Diverse Leadership Is There a Danger of Mirroring in Reverse, i.e., Do We Simply End up with Another Idealised and Dominant Leadership Hegemony?

The case for a shift towards the DLMOX paradigm, as we have advocated, starts at the sociocultural levels by problematising the idealised, universalised leadership canon with its (colonial) knowledge hierarchy and its pedagogical processes that legitimate and maintain such a leadership monoculture. However, in advocating, we also always need to take great care not to set up unhelpful dichotomies such as *Western hegemonic bad* vs. *non-Western diverse good*. Jimenez-Luque [73] captures this tension when he argues that

“it is important to clarify that the critique of the hegemony of Western knowledge does not imply that Western epistemology is not valuable. A critical and decolonial perspective means that although Western knowledge in general, and Western theories of leadership in particular, are central to understanding [sic] the complex phenomenon of leadership, they (1) are not the only ones, (2) only represent a local perspective, and (3) need to acknowledge their position of power and privilege regarding other cultures as a result of colonialism and the epistemicide.”

Simply repudiating the Western hegemonic paradigm outright, without understanding it, may unwittingly create a new idealised diverse leadership hegemony, whereby the Western epistemology is summarily dismissed in favour of all that is non-Western. As such, there is an inherent tension and a risk of mirroring in reverse the universalising colonial logic of Western leader-centrism [72] that we are here trying to advocate against. The ramification of a shift towards diverse leadership for organisations is that ongoing collective discussion amongst leaders and members, of all social identities and lived experiences, of the prevailing leadership logic and its effects will be a critical antecedent to the process of rethinking leadership. Rethinking leadership certainly requires unlearning and decolonising [72,73], though great care should be taken in navigating this choppy passage. This is all easier said than done, but the following (sub)questions might be a useful starting point.

### 4.3. If Dominant Western Leader-Centric Approaches Are Increasingly Inadequate and Unresponsive to RAPID Change, Then How Do We Begin to Rethink Leadership?

Implicated in any rethinking enterprise is understanding the views of leaders and members. Understanding the significance of paying attention to all constituents, Chin, Desormeaux, and Sawyer [74], for example, ran a diverse leadership summit that was predicated on the following questions:How do you view leadership?How is your exercise of leadership influenced by the context in which you lead, the multiple dimensions of your identity, and your lived experiences associated with culture and minority status?How do you project the kind of leadership needed for the future, given the rapid change, growing diversity, and increased globalisation in society?

The participant responses gave rise to the following four competencies:Leveraging personal and social identities;Utilising a global and diverse mindset;Leveraging community and organisational contexts;Promoting a diversity-supportive and inclusive climate.

In turn, each competency was underpinned by detailed sets of dimensions (see [74]). The significance is that rethinking leadership should be collective and socially constructed rather than prescribed or mandated. The ramification of the positional leader ‘going it alone’, by failing to include all constituents is that the prevailing leadership orthodoxy simply gets reproduced under a new name. Rethinking leadership from a global and diverse perspective should start by asking leaders and members, across dimensions of social identity and lived experience, some foundational questions about (a) the nature of leadership, (b) the role of context, and (c) the congruity between the demands of our globalised world and how leadership is construed, enacted, and evaluated. Such rethinking, according to Chin [70], starts with unlearning and might also include a discussion of the following and their potential ramifications:What and whose knowledge, practices, and behaviours are privileged, propagated, and protected? In what and whose name is the prevailing leadership paradigm in service?What epistemologies and knowledge systems are missing or have been banished from the dominant picture? Is there room for expanded conceptualisations of leadership?Who is missing from the leadership picture, who are the unfavoured categories, and who gets to lead?

However, when asking these questions, there will naturally be degrees of scepticism, fear, and rejection, especially from the dominant in-groups. This brings us to the next (sub)question.

### 4.4. What Are Some of the Challenges of Shifting towards Diverse Leadership, Especially for the Command-and-Control-Style Leader and for Those Whom They Lead?

Naively expecting dominant in-groups, who may have vested interests in maintaining the status quo, to change their leadership thinking and practice will invariably be accompanied by scepticism and resistance. One obvious implication of shifting towards diverse leadership will be fear of losing legitimacy, i.e., leader identity, power, influence, authenticity, and credibility. Diverse leadership could be (mis)construed as soft, indecisive, and endlessly acquiescent, and misattribution has real implications if left unchecked. Recognising this tension, Chin and Trimble’s notion of the diverse leader adopting an ‘affirmative paradigm’ [3,70,71] is significant because they contend that diverse leaders are, amongst other qualities, called to be collaborative, collectivist, altruistic, and socially just but at the same time to be unabashed by accepting the title of leader, having the courage to make decisions and to take risks, and embracing difference rather than apologising for it.

In the same vein, tensions may arise for members who hold tight to ingrained cultural expectations about what and who their leaders should be and about what constitutes leader legitimacy and effectiveness across societal, organisational, and individual levels (see [59,64]). The potential ramifications of orthodox-style leaders modifying their leadership styles to account for context are that members become concerned about their leader’s authenticity [3,70]. Therefore, pre-empting leader and member expectations and concerns regarding change should be a collective endeavour that implicates the organisation across its climate, core/curriculum, and composition so that individuals or teams are not left to navigate these tensions ad hoc and alone.

As a case in point, take an organisation that has recruited and promoted leaders with more diverse social identities than the dominant in-group. Subsequently, it turns out that these diverse leaders simply end up leading more like the command-and-control in-group who recruited and promoted them in the first instance (see [69,72]). This is an example of where the ambition to become more compositionally diverse has not necessarily been matched by a diversity-supportive climate and core/curriculum. This organisation may look more representationally diverse but is far from culturally competent or equitable, and this may create tensions amongst constituents with respect to both the inclusion strategy and the newly appointed leader’s authenticity. One recommendation that Chin [70] proposes is to ‘leverage social identities, lived experience and the organisational climate’ by establishing what she calls ‘interdisciplinary intercultural partnerships’ as a way of advancing greater cultural competence, which might start by exploring the aforementioned scepticism, fear, and resistance.

### 4.5. All This Collaboration and Adapting to Context Takes Time—Is It Worth It?

There is no escaping the fact that it will take time for leaders to gain a diversity of perspectives by paying attention to members across organisational levels and across dimensions of social identity. While collaboration and a collectivist approach demand more time and energy, without this investment, innovation arguably remains stifled. Where leaders take the time to obtain multiple perspectives, including dissenting voices and contrarian views—and all things being equal—diverse leadership has the potential to mitigate against ‘group think’ [3,70], whereby individuals acquiesce in homogenous groups to reconfirm and reproduce each other’s preferences, biases, and decisions. The ramification of this for organisational climate and curriculum is to deliberately promote, practice, and recognise divergent thinking across functional levels and member identities so that innovation can flourish. What might emerge if there was less concern with hierarchical lines of command and correspondingly less opportunity for groupthink? Difference makes a difference when it comes to collaboration and innovation, and this is what diverse leadership can bring [3,70,71]. But how should we prepare leaders and organisations to collaborate with diverse members, to adapt to diverse contexts, and to be socially just in doing so?

### 4.6. How Should We (a) Prepare Leaders for Diverse Leadership and (b) Evaluate Them?

In chapters 8 and 9 [3], Chin and Trimble consider implications and dilemmas for leadership development at individual and organisational levels, as well as aspects of evaluation within their paradigm (leadership effectiveness assessed across quantitative, qualitative, and evaluative data). While a fuller explication of these chapters is beyond the scope of this section, the authors identify three ‘applications’ for training culturally competent leaders:Leader self-awareness: Self-assessment;Leader skills: Developing cultural fluency;Leader knowledge: Dimensions of cultural difference.

They then present several dilemmas implicit in the development of diverse leaders, such as self-awareness and identity, expectations and image management, negotiation, intercultural communication, collaboration and decision-making, problem-solving and managing conflict, work culture, and team-building (global teams). The central point to be made here is that for leaders to be culturally competent and responsive in the adaptive spaces between diverse member composition, organisational climate, and member expectations, they will be required to adapt their styles across contexts. The fundamental challenge and tension then are to find ways of becoming more ‘practically wise’, which Jones and Hemmestad [75] describe as someone who displays self-awareness, discernible judgement, intuitive situational literacy while attuned to emergence, context, and ethical imperatives (*phronesis*). As such, the wider ramification of this rethinking is that leadership development and evaluation of leadership is no longer simply about the production of something that requires predetermined outcomes and assertive top–down influence (*poiesis*).

Although the above discussion of implications is far from exhaustive or prescriptive, our aim is to explore the significance, tensions, and ramifications involved in a paradigm shift towards diverse leadership for sports leadership and leadership beyond.

## 5. Conclusions

Sports leadership, like other areas of human endeavour, operates in increasingly heterogeneous global and diverse contexts through exchanges between leaders and members with ever-diverse social identities and lived experiences. The purpose of this article, in line with the objective of this Special Issue, is to advocate for ways of ‘rethinking leadership’. In this article, we challenge the Western orthodoxy of leadership because it is overwhelmingly leader-centric, ethnocentric, gender-biased, and representative of a narrow range of social and cultural identities. As such, we advocate for a diverse leadership paradigm that is more socially just, altruistic, collaborative, and inclusive of *all* its members across dimensions of social identity and across organisational levels—one that is ‘culturally competent’—even if there is a lack of coherence as to what constitutes cultural competence. Our contribution to this Special Issue highlights that diverse leadership can only be realised if we are conceptually clear on what cultural competence is and how it can be collectively constructed and embedded throughout the organisation’s climate, core/curriculum, and composition. Gaining clarity around cultural competence is important because it is a catalyst for diversity and enhances the possibility that a paradigm shift towards diverse leadership can gain momentum. We invite practitioners and researchers in sports leadership and beyond to consider the potential of diverse leadership as an integrative and yet fluid paradigm. Rethinking leadership and the development of our leaders for a human future together has never been more necessary and urgent, and diverse sports leadership has an important part to play.

## Data Availability

Data are contained within the article.

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
