# Peer review of "It Is Leadership, but (Maybe) Not as You Know It: Advocating for a Diversity Paradigm in Sports Leadership and Beyond"

_behavsci, 2024, doi:10.3390/bs14100860_

Round 1

Reviewer 1 Report

Comments and Suggestions for Authors

The subject matter of this article is very interesting and relevant, particularly with regard to its perspective on leadership within the realm of sports, which highlights critical facets of the topic. However, although this is a very interesting story, I am not sure whether a thorough analysis and conclusive insights can be derived from a singular case study. The recommendation for future work would be to conduct actual research on this subject which could provide valuable opportunities to establish more generalizable conclusions.

Author Response

Dear Reviewer

Thank you for your informative comments.  Based on these, and the summation made by the Editor of all three reviewers’ comments, we have extensively reworked the original article submitted for review. Therefore it is not possible to respond to your review point by point. We acknowledge the confusion caused by including the narrative into the original submission. This resulted in all reviewers requesting more information about methodology etc. To resolve this confusion we have removed the narrative about Frankie and any discussion in the introduction about narrative. In the resubmission we have now made it very clear that this is a conceptual article that advocates for us to ‘rethink leadership’. In the introduction we made it clear how the article has relevancy to a broad readership and  the purpose and significance of the article more made more prominent. In the revision we have strengthening the discussion of the literature and  the diverse leadership paradigm. A major reworking of the resubmitted article is based on the Editor’s request that we expanded the discussion of implications for leadership development. We have done this and it now comprises nearly 25% of the article.

 We hope that our extensive reworking of the article satisfies any concerns you had about the original submission. Moreover, we would like to thank you for challenging us to become more precise in our thinking and the way we present our arguments regarding why we think it is important to rethink leadership development. 

Warmest regards

Tania and Gary

Reviewer 2 Report

Comments and Suggestions for Authors

Dear authors,

thank you for the opportunity to review the article with this interesting topic, relevant in the sports world. 

However, before I am able to recommend the article for publishing, you will need to improve its quality. Here are my remarks that need to be addressed by you in the revisions: 

-       Improve the title’s clarity and understandability, 

-       There is no table or figure in the article, which makes it less interesting for the readers and makes the understanding harder – you can add a summarizing table to compare the literature sources analysed (their differences, similarities,…) and either your own scheme depicting the structure of the topic (elements and their relationships) in the introduction or you can present the summary of all your results in a mind map form at the end of the results section, 

-       Do not include any references in the abstract, 

-       Add a description of your methodology and a highlight of the results in the abstract, 

-       Do not use long citations in the introduction – create your own text based on the references, with the same main points, 

-       Avoid informal language (e.g. „flavour“, „speaks for itself“) 

-       Abbreviations need to be explained in full at the first occurrence, 

-       The overall structure needs improvement – adhere to standard academic writing (1 introduction, 2 theory, 3 methods, 4 results, 5 discussion, 6 conclusions), 

-       You need to create the whole part describing your methodology – aim, data (even if they are just secondary – cases, etc.), methods, 

-       Your results chapter can include a comparison of multiple case studies with your original findings stemming from it, 

-       You also need to create a discussion chapter – confronting your findings with those achieved by other authors all over the world, 

-       Within the conclusion, add specific recommendations for the stakeholders relevant to the analysed topic, 

-       Also, at the end of the conclusions describe research limitations and future directions. 

I hope that these comments will help you increase the article’s quality sufficiently so the revised version will meet the journal’s standards. 

Have a nice day. 

Reviewer

Comments on the Quality of English Language

The informal style needs to be changed to formal, and the title needs more clarity. 

Author Response

(The authors gave the same response as above.)

Reviewer 3 Report

Comments and Suggestions for Authors

Dear authors,

I enjoy reading the manuscript. The paper is very well written and easy to read. It also sounds like an important subject to explore in sports leadership. Frankie's story is very intriguing. In the meantime, I also have a few observations to share with you and hope they are helpful. 

1. Research question: you spent over two pages for the introduction. Yet, after reading this part, I am not sure I understand your research question or the rationale behind the study and its contributions (i.e., what we can learn from reading your paper?). You use the phrase "this article is to contribute to the discussion on paradigm shifts" (line 171) without elaborating on what and how you contribute to the conversation with your study. 

2. Methods: What is also not clear is the methodology. I could not really tell if this would be a conventional qualitative study. You have certain components of the method but the process and writing are not conducted rigorously by adhering to the methodology. The main section of the paper reads like a book review, mixed with a reporter's interview story. 

3. Literature review: You spend a lot of space reviewing a particular work (Chin & Trimble). One or two works might not be enough to build up the state of the literature or the subject field. More importantly, a literature review should be a synthesis of the literature and articulated in your own voice. Readers need to see, from your vantage point, what the literature is saying about your research questions, why that is significant for the paradigm shift conversation, and what is missing yet in the literature. This paves the way for you to say here is what our study can help to build the bridge and address the gap to help better understand the phenomenon. 

Simply book-reviewing one or two particular works without emphasizing how they are related to your research questions and contributions also creates a misalignment between your "literature review" and the methodology.

4. Vignette and analysis: Chin & Trimble's review covers a lot of perspectives. But your Frankie's story is comparatively thin. It is a very brief interview/self-narrative. As a reader, I do not see the story either exemplifying Chin & Trimble's framework or shedding light on the shift at the paradigm level. 

It is also not clear how the interview was conducted or if it followed the methodology design principles such as what type of interview is this, structured or semi-structured? 

Author Response

(The authors gave the same response as above.)

Round 2

Reviewer 1 Report

Comments and Suggestions for Authors

Dear authors,

Thank you for taking into account my suggestions and reformulating the paper.

As far as I'm concerned, everything is fine now.

Best regards!

Author Response

Thank you for assisting us to improve this article.

Best regards

Reviewer 2 Report

Comments and Suggestions for Authors

I have checked the revised version of the article and the authors‘ responses.

After the changes and their explanation, I now understand the nature of the paper better, being a conceptual paper, not a typical research paper including results of data analysis.

Thus, if this works for your special issue, I can recommend the paper for publishing (after the formal editing provided by the journal).

Hope this reply helps you with your final decision.

Comments on the Quality of English Language

NA

Author Response

Thank you for assisting us to improve our article.

Warmest regards

Reviewer 3 Report

Comments and Suggestions for Authors

Dear Authors, 

Your passion for the issue is contagious and I also appreciate your tremendous effort in revising this manuscript. I understand that you have changed the aim and focus of this paper to develop a conceptual paper. Here are my observations of this revision:

1. Research question: I think one key issue of this revision is still the lack of clearly framed research questions, which should be developed on the ground of identified gaps in the literature. For instance, your writing centers around the phrase "rethinking leadership" but you never really explain what current thinking of leadership is other than a broad terminology. And if you are taking on a paradigm (which is a huge undertaking), it will be helpful to elaborate what the problems are in theoretical sense. It is pretty difficult to follow because of these issues right off the bat. 

2. Conceptual Paper: I am not sure how to interpret the way you structure the paper. You say you want to rethink the entire paradigm of leadership. Do you intend to propose a new theory or a new paradigm?

 A conceptual paper, as I see it, would probably take on two formats, one is a conventional theory paper, which builds up propositions theorizing the relationships among core constructs. Another commonly seen format is a narrative or conceptual review of a literature, which should take on a systematic methodology. 

I am not seeing any of the basic elements of a conceptual paper in your manuscript. Instead, your paper reads more like a commentary piece for practitioners, or part of leadership development program planning.

3. I think it is also telling that your references are mostly from general publications for business instead of academic outlets. 

In conclusion, I am wondering if your manuscript is a good fit for the journal.

Comments on the Quality of English Language

Minor English editing is needed.

Author Response

Dear reviewer,

Thank you for your feedback which has assisted us to improve the article. Below we have responded to your comments from the second review (see italicised sections) 

  1. Research question: I think one key issue of this revision is still the lack of clearly framed research questions, which should be developed on the ground of identified gaps in the literature. For instance, your writing centers around the phrase "rethinking leadership" but you never really explain what current thinking of leadership is other than a broad terminology. And if you are taking on a paradigm (which is a huge undertaking), it will be helpful to elaborate what the problems are in theoretical sense. It is pretty difficult to follow because of these issues right off the bat.

In the abstract we do explicitly pose the general question we are concerned with, which aligns with the non-empirical 'rethinking' nature of this paper. We believe that 'Rethinking' is sufficiently invitational, open, and capacious to allow for a broad question that orients rather than prescribes. Our work is not empirical, although it draws on multiple empirical sources. Additionally, we present a number of emergent questions in the second half of the paper that align with the question we pose in the abstract, together with the declared purposes of the paper.

We have attempted throughout the paper to problematise much current leadership thinking (orthodox universalised leadership approaches / Western leadership canon / idealised leadership  / leadercentism / etc.,). The terminology we use in the article is purposefully  broad and it is our preferred approach in this advocacy paper given our understanding of the context of the journal's readership. However, you'll note that we refine matters in the latter third of the paper. 

In this advocacy paper we align with the notion that research does not only exist to ask specific questions, or to solve specific problems; research should also cause some problems where nobody saw problems before (Siegel and Biesta, 2022)

  1. Conceptual Paper: I am not sure how to interpret the way you structure the paper. You say you want to rethink the entire paradigm of leadership. Do you intend to propose a new theory or a new paradigm?

'Rethinking' does not mean 'replacing' (see our Implications section where we caution against this), and therefore our aim is to advocate for expanded construals of leadership. Diverse leadership as a paradigm, is itself at least a decade in existence. We are simply drawing on this pre-existing work to contribute to this ongoing paradigm shift, and to advocate for rethinking leadership accordingly, specifically in the field of sport, in which we work.

Ours is a stated advocacy paper that seeks to contribute to rethinking leadership, and so we specifically draw on what Chin and Trimble themselves call their 'DLMOX paradigm'. We are advocating for, and contributing to, a paradigm which already exists in general leadership studies i.e., the DLMOX paradigm (Chin and Trimble, 2014) . Our specified context is sport leadership and leadership in sports coaching. A diverse leadership paradigm would be new to our field, but not necessarily elsewhere.

 A conceptual paper, as I see it, would probably take on two formats, one is a conventional theory paper, which builds up propositions theorizing the relationships among core constructs. Another commonly seen format is a narrative or conceptual review of a literature, which should take on a systematic methodology. 

Throughout the paper we have attempted to explicitly and sequentially build up propositions and connections within and between the four sections, as supported by general and context specific literature and sources.  

I am not seeing any of the basic elements of a conceptual paper in your manuscript. Instead, your paper reads more like a commentary piece for practitioners, or part of leadership development program planning.

Our declared advocacy approach is predicated upon a number of interrelated concepts which we have presented, explicated and commented upon (sociocultural, social construction, social identity, cultural competence, diverse leadership).

The focus of this Special Edition is on 'Rethinking Leadership Development' and so we have attended to the leadership development dimension.

The editors' asked that a dedicated section on 'implications for leadership development' to be included, and so we have also attended to that.

  1. I think it is also telling that your references are mostly from general publications for business instead of academic outlets.

We draw on academic and peer reviewed work throughout. This academic work is chosen because of its relevance to business, education, health, society and to sport leadership.  We explicitly draw on a range of literature in an attempt to be sensitive to the broad readership of this journal rather than drawing more exclusively on our specialised field of sports coaching and coach development in sport.

While our declared context is sport leadership and sports coaching we have built up our advocacy case by drawing on sociocultural and pedagogical perspectives. The Deloitte report we cite is business focused, yet its title has much wider ramifications - that is why we chose it. With respect to the work of Chin and Trimble's and colleagues, who we reference in parts three and four of the article, all of these authors are academics publishing in academic outlets. The three videos cited are of Chin and Trimble separately presenting ideas on diverse leadership, and the role of cultural competence in diverse leadership.

Regards